# Machine Learning-Based Feature Selection and Classification for the Experimental Diagnosis of *Trypanosoma cruzi*

**Nidiyare Hevia-Montiel** [1,*], **Jorge Perez-Gonzalez** [1], **Antonio Neme** [1] and **Paulina Haro** [2]

1   Unidad Académica del Instituto de Investigaciones en Matemáticas Aplicadas y en Sistemas del Estado de Yucatán, Universidad Nacional Autónoma de México, Mérida 97302, Yucatán, Mexico; jorge.perez@iimas.unam.mx (J.P.-G.); antonio.neme@iimas.unam.mx (A.N.)
2   Instituto de Investigaciones en Ciencias Veterinarias, Universidad Autónoma de Baja California, Mexicali 21386, Baja California, Mexico; paulina.haro@uabc.edu.mx
*   Correspondence: nidiyare.hevia@iimas.unam.mx

**Abstract:** Chagas disease, caused by the *Trypanosoma cruzi* (*T. cruzi*) parasite, is the third most common parasitosis worldwide. Most of the infected subjects can remain asymptomatic without an opportune and early detection or an objective diagnostic is not conducted. Frequently, the disease manifests itself after a long time, accompanied by severe heart disease or by sudden death. Thus, the diagnosis is a complex and challenging process where several factors must be considered. In this paper, a novel pipeline is presented integrating temporal data from four modalities (electrocardiography signals, echocardiography images, Doppler spectrum, and ELISA antibody titers), multiple features selection analyses by a univariate analysis and a machine learning-based selection. The method includes an automatic dichotomous classification of animal status (control vs. infected) based on Random Forest, Extremely Randomized Trees, Decision Trees, and Support Vector Machine. The most relevant multimodal attributes found were ELISA (IgGT, IgG1, IgG2a), electrocardiography (SR mean, QT and ST intervals), ascending aorta Doppler signals, and echocardiography (left ventricle diameter during diastole). Concerning automatic classification from selected features, the best accuracy of control vs. acute infection groups was 93.3 ± 13.3% for cross-validation and 100% in the final test; for control vs. chronic infection groups, it was 100% and 100%, respectively. We conclude that the proposed machine learning-based approach can be of help to obtain a robust and objective diagnosis in early *T. cruzi* infection stages.

**Keywords:** machine learning; feature selection; multivariate analysis; classification; Chagas disease; *Trypanosoma cruzi*; echocardiography; electrocardiography; doppler; ELISA

## 1. Introduction

The clinical diagnosis of a disease is a complex process since there are several factors and symptoms that must be taken into account and analyzed and integrated by clinical expertise. The use of machine learning techniques has proven to be a valuable tool in the diagnosis process [1]. Machine learning offers medical personnel with techniques and methods that allow the selection of appropriate variables, a rational classification of patients based on their stage for a given illness, and an accurate prediction of disease progression. These computational methods and algorithms can help to obtain a robust and objective clinical diagnosis. In addition, for some clinical diagnoses, more complex situations can occur, such as diseases with long asymptomatic periods, as is the case for Chagas disease (CD) patients.

The World Health Organization (WHO) recognized, in 2010, CD (American trypanosomiasis) as one of the 17 neglected tropical diseases in the world [2,3], and it is considered the third most common parasitic infection in humans, only behind malaria and schistosomiasis [4]. CD is endemic in 21 countries in Latin America and, due to migration, cases in Canada, United States of America, European, African, Eastern Mediterranean and Western

Pacific countries have also been identified. This disease affects around 7–8 million people, with 75 million people at risk of infection [5]. Between 30–40% of the affected population are prone to developing cardiovascular, gastrointestinal, neurological problems, or all of the above cases due to the disease [3].

CD is caused by the protozoan *Trypanosoma cruzi* (*T. cruzi*). Infection occurs majorly by vectorial transmission, but other transmission mechanisms such as through blood transfusion or organs transplant, vertically (mother to child) [6], orally, and during laboratory accidents are also possible [7]. CD clinically manifests in two phases: acute and chronic. The acute phase in humans lasts two to four months with high parasitemia (presence of parasites in the blood stream), active division on nucleated cells, and clinical presentation can be asymptomatic or symptomatic. In the chronic phase, *T. cruzi* is harbored into certain tissues cells, with tropism in muscular cells, most commonly heart and digestive organs [8]. The chronic phase can be asymptomatic, and it is diagnosed by detecting antibodies against *T. cruzi*. After 10–30 years after the infection, symptoms begin to emerge, which may result in severe heart disease, heart failure and sudden death [9].

Since CD affects several organs and tissues, the presence of *T. cruzi* can be detected by a systematic inspection, where several diagnostic tests can be performed in order to detect or study the disease progression of the infected individual. The stage of the infection can be estimated, we hypothesize, from these diagnostic tests and by applying a handful of relevant classifiers, to be discussed in the next sections. Due to the wide range of the referred signals, we will now briefly describe them as well as their biological sources.

The CD diagnosis in the acute phase is based on positive parasitological tests. A common technique used in the detection, diagnosis and/or monitoring of patients with CD is the Polymerase Chain Reaction (PCR), which is a technique used in molecular biology able to detect DeoxyriboNucleic Acid (DNA) of *T. cruzi* [10,11]. For patients in the chronic phase, the preferred method is the Enzyme-Linked Immunosorbent Assay (ELISA), usually used to detect antibodies by a visible color change or fluorescence from quantitative or qualitative measurements based on colorimetric reading [12,13]. If the symptomatic chronic phase is suspected, heart disease and myocardium damage can be diagnosed using electrocardiography (ECG) or echocardiography (ECHO), which also provides structural and functional heart information [14,15].

Non-invasive or minimally invasive diagnostic methods are sought to carry out the diagnosis of a disease; thus, diagnostic imaging techniques have an impact as a medical diagnostic tool since they allow the visualization of internal organs and at the same time, make it possible to inspect their function. Medical images have become an essential component in the *in vivo* follow-up of pathological changes associated with *T. cruzi* infection studies. ECG serves to evaluate the electrical activity of the heart by a visual representation of the time-voltage heartbeat relationship and, simultaneously, allows the analysis of the depolarization (heart muscle cells activation process) and the repolarization (return of cardiac cells to their resting state after depolarization) of the different heart anatomical regions in order to detect functional changes in the heart. Cardiac ultrasound or ECHO provides real-time images of the heart structure and functional information. Spectral Ultrasound Doppler provides information of hemodynamic function, allowing to determine different parameters of blood flow measurement [16,17].

## 1.1. Previous Work on Diagnosis Techniques and Clinical Findings for T. cruzi

Several contributions have reported the relevance of diagnostic techniques such as ECG and ECHO for the *T. cruzi* case. In Yacoub et al. [18], a protocol is described for patients suffering from CD using non-invasive methods aiming to detect early heart damage. There, the authors analyzed variables extracted from ECG and ECHO from 133 patients. It was shown, following a statistical analysis, that the myocardial performance index, which is a parameter obtained from the Doppler spectrum, and the ratio between the thickness of the posterior wall to the left ventricular cavity, obtained by ECHO, are relevant characteristics for early detection of myocardial damage in asymptomatic patients with CD.

Viotti et al. [19] present a study with 849 patients based on ECG acquisitions, radiography and 2D ECHO, performing both a univariate and multivariate analysis for clinical events and mortality among four groups of patients. Their results identify the variables related to the left ventricle systolic dimension and the ejection fraction as relevant predictors of clinical events as well as mortality. These authors determined that ECHO analysis was important to characterize and determine the prognosis in patients without heart failure in the chronic stage. Valerio et al. [20] reported a study to determine clinical abnormalities from ECG and ECHO acquisitions in CD infected patients at the diagnosis time, in which 49 patients presented abnormalities significantly associated with ECG and/or ECHO parameters ($p = 0.038$) in patients newly diagnosed with the infection.

Immunological tests of CD have shown their relevance as diagnostic tools. For instance, Ferrer et al. [21] present a classification method sustained on a three-way scheme comparison: i. CD patients in the acute and chronic phase, ii. patients with other diseases and CD, and iii. CD healthy subjects, all based on immunological and molecular tests. Immunological techniques reported a precision of 69.2% in acute phase patients and 95.2% in chronic phase patients, while molecular tests obtained a precision of 79.5% in acute phase patients and 23.8% in chronic phase patients, performing an assessment of concordance of the results.

Several additional works have been conducted to monitor the disease evolution, such is the case of Oliveira et al. [22], who carried out an experimental study to detect anomalies in the movement of the left ventricular wall from an animal model in the chronic stage with cardiomyopathy. The animals were evaluated *in vivo* by the global and segmental left ventricular systolic function based on ECHO evaluation. Statistical analysis showed a negative correlation of the left ventricular ejection fraction in infected animals. Santos et al. [23] carried out an experimental study to determine, through ECHO techniques, the medical treatment efficacy to reduce the left ventricle diastolic dysfunction, where the left ventricle diastolic function was measured based on the mitral and pulmonary inflow pattern by Doppler images. These authors presented a statistical analysis based on Pearson's linear correlation of diastolic function parameters from Doppler images, showing that the development of diastolic function is correlated with cardiac lesions.

### 1.2. Machine Learning in Chagas Disease Classification

Machine learning has been applied in the study of CD in humans. Teles et al. [24] evaluate the potential use of machine learning and the automatic selection of attributes in discrimination of individuals with and without CD, based on clinical (ELISA and indirect immunofluorescence) and sociodemographic data (gender, age, education level, kind of housing, therapeutic treatment, handling or contact with triatominae, earlier diagnosis, and history related to cardiovascular and digestive systems), where automatic attributes selection methods (forward selection and backward elimination) and classification algorithms (Genetic Neural Network Multilayer Perceptron (MLP) and Linear Regression (LR) were implemented. Silva et al. [25] studied the relation between heart rate variability (HRV) and ECHO in a population of patients with CD, with the hypothesis that HRV can be used to predict numerical and categorical echocardiographic parameters. They calculated twenty-seven HRV indices, and eight parameters were obtained from the transthoracic two-dimensional echocardiograms; machine learning models were trained to predict the echocardiographic parameters taking into account the HRV indices as inputs. Silva et al. trained separated models for each echocardiographic parameter, where each parameter (numeric or categorical) was considered a separate problem. Four different machine learning algorithms were used in their study: Random Forest (RF), MLP, *K*-Nearest Neighbors with the KStar estimator, and Support Vector Machine (SVM) using the sequential minimal optimization. Escalera et al. [26] present an analysis based on the features extraction from 10 min of high-resolution ECG recordings of 107 patients with CD of the QRS complex to perform a multi-classification system able to learn the level of damage produced by the disease, focused on error-correcting output codes as a general framework to combine

binary problems to address the multi-class problem, and based on the error correcting the bias and variance errors of the base classifiers; this work presents a comparison of the ECOC algorithm with Discrete AdaBoost and Linear Support Vector Machines (LSVM) algorithms for the classification of 107 patients according to the cardiac damage degree. Asl et al. [27] present an arrhythmia classification method from the heart rate variability (HRV) characteristics of 47 ECG recordings from a mixed population; they performed a features reduction by General Discriminant Analysis during the training of an SVM algorithm, thus obtaining a greater precision compared to the use of all originally extracted ECG variables.

As described in the previous paragraphs, the detection of the CD and the determination of the clinical stage represents a challenge. Therefore, the correlation analysis of variables based on different diagnostic methods often performed during *T. cruzi* infection studies, and their association according to the clinical stage of the infection in a murine model can be useful to establish a diagnostic approach criteria.

In this paper, a novel pipeline that integrates temporal data acquisition from four modalities, multiple features selection analyses, and an automatic classification strategy of *T. cruzi* infection model is presented. The variables of a murine experimental model of *T. cruzi* infection were obtained from: ECG signals, ECHO images, Doppler spectrum, and ELISA antibody titers. These modalities are related to the presence of certain antibodies and abnormalities detection in cardiovascular functionality during the infection. Afterward, a univariate and multivariate analysis of the multimodal feature relevance was performed. Finally, a set of supervised classifiers was proposed, fed with variables from each modality individually and different subsets combining the four modalities studied. The task is to automatically classify *T. cruzi*-infected animals from the murine experimental model for acute phase (control vs. infected), chronic phase (control vs. infected), and general infection groups (acute phase + chronic phase vs. controls).

## 2. Data Acquisition and Methodology

This section presents the experimental murine model of *T. cruzi* infection and the multimodal diagnostic techniques (ECG, ECHO, Doppler, and ELISA) for the acquisition of variables involved in the diagnosis of *T. cruzi* and heart damage caused by the presence of the parasite.

### 2.1. Experiment Description

The experimental infection model and data acquisition from 72 mice was carried out in the Parasitology Laboratory of the Centro de Investigaciones Regionales Dr. Hideyo Noguchi at the Universidad Autónoma de Yucatán (CIR-UADY), Mexico. A murine model of *T. cruzi* infection in the acute and chronic phase was performed [28–30]. The experiment was approved by the ethical committee of the CIR-UADY (CIRB-006-2017). The animals were handled according to the Guide for the Care and Use of Laboratory Animals (eighth edition) [31]. An experimental murine model was implemented for acute and chronic stages of *T. cruzi* infection. The experiments were performed in duplicate, 3 control and 3 infected animals were analyzed par day during the acute and chronic stage, and the infection model experiment was repeated to complete 6 control and 6 infected animals on each day of study. The temporal data acquisition was obtained at different post-infection days (12 mice par day randomly selected: 6 control mice and 6 infected mice), where all multimodal variables of each animal were acquired on the same day.

For the acute phase, a total of thirty-six 6–8-week-old healthy female ICR mice were considered, divided into control (18 mice) and infected (18 mice) groups. The mice of the control group received physiological saline administered intraperitoneally. For the infected group 1000 blood stream trypomastigotes of the H1 *T. cruzi* strain, previously isolated from a human case in Yucatán, Mexico, were also administered intraperitoneally. A multimodal data sampling period was conducted in 6 infected and 6 control animals at 15, 25 and 35 days post-infection.

For the chronic phase, thirty-six 6–8-week-old healthy female ICR mice were considered, divided into control (18 mice) and infected (18 mice) groups. The mice of the control group received physiological saline administered intraperitoneally, while the mice of the infected group were infected with 500 blood stream trypomastigotes of the H1 *T. cruzi* strain. Multimodal data were acquired on 6 infected and 6 control animals each 60, 90 and 120 days post-infection.

The development of infection for the acute and chronic stage was confirmed and monitored, in order to validate the infection, by counting peripheral blood parasites by microscopy using a Neubauer cell count every 5 days until 35 (in the acute stage) and every 30 days until day 120 (in the chronic stage). Mortality was recorded daily.

*2.2. Data Acquisition*

Multimodal data acquisition was performed based on four diagnostic techniques: ECG, ECHO, Doppler, and ELISA, where the multivariables considered for the analysis are summarized below. All multimodal samples were acquired over 24 h during the post-infection day for each study subject, both for control and infected groups in the acute or chronic phase.

Mice cardiac activity was recorded with non-invasive ECG equipment (Mouse Specifics Inc., Quincy, MA, USA) for conscious animals using lead II. A series of 20–30 consecutive ECG complexes were obtained from each animal. The ECG signals were analyzed using software (EzCG Analysis Software package, Mouse Specifics Inc., Quincy, MA, USA), and fourteen variables were considered related to PQ, QTc, PR, QT, RR, ST Intervals, as shown in Figure 1, in addition to QRS Complex, Heart Rate (HR), HRV, QTc dispersion, Cardiac Variability (CV), SR mean, and R amplitude mean. To determine the QTc, the equation consists of a modified Bazett equation used in humans [32,33], as is shown in Equation (1), for an approximate duration of the RR interval in mice of 100 ms:

$$QTc = QT/[\sqrt{RR}/100] \tag{1}$$

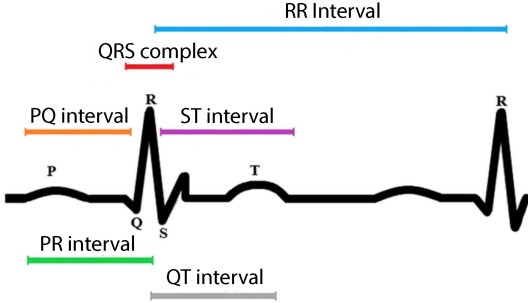

**Figure 1.** The P wave (atrial depolarization) and QRS complex (ventricular depolarization); PQ interval (beginning of the P wave to the beginning of the QRS complex); PR interval (beginning of the P wave to the peak of the QRS complex); ST interval (ventricular repolarization); QT interval (ventricular depolarization and repolarization); and RR interval (time between ventricular depolarization peak). SR (mean amplitude of the signal measured from each signal minimum (S) line to the peak of the R-wave.

To obtain ECHO images, the mice were anesthetized using inhaled anesthesia (Patterson Scientific, Waukesha, WI, USA) at an induction dosage of 3% isoflurane and 0.5 L/min $O_2$ using a chamber and maintained through a face mask with a dosage of 1.5 to 2.5% and 0.5 L/min $O_2$. ECHO was performed using a 22 MHz lineal transducer and Mylab Seven equipment (ESAOTE S.P.A.®, Florence, Italy). Left ventricle (LV) images were obtained in long and short axis views, in B and M mode. A total of 5 variables were obtained: HR, LV diameter at the end systole (LVs) and diastole (LVd), systolic function was evaluated by

calculating fractional shortening (FS), and the ejection fraction (EF) was calculated by the next Teichholz Equation (2):

$$Vol = 7D^3/(2.4 + D) \tag{2}$$

where *Vol* is the LV volume and *D* is the LV diameter [34,35].

Under anesthesia, blood flow acquisitions were obtained using a Doppler system for mice (INDUS Doppler System). Signal at the level of the Mitral Valve (MV) and Ascending Aorta (AO) were acquired using a 10 MHz transducer, and a 20 MHz transducer was employed for the acquisition at the level of Abdominal Aorta (AbAO). Anatomical marks and spectral signal characteristics were used to identify each structure. From the Doppler spectral signals, which are composed of frequency changes generated by blood flow changes, 45 variables were extracted related to parameters of heart rate, peak velocity, flow velocity, pulsation, and acceleration [36], as is shown in Figure 2.

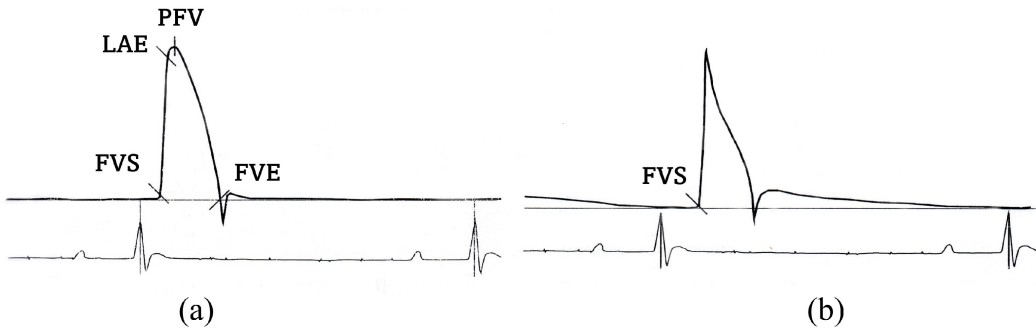

**Figure 2.** Doppler spectral variables. (**a**) Ascending aorta: FVS (Flow Velocity Start), LAE (Linear Acceleration End), PFV (Peak Flow Velocity), FVE (Flow Velocity End), and (**b**) Abdominal aorta: FVS.

For antibodies detection against *T. cruzi*, an ELISA test was conducted. A blood sample was obtained by cardiac puncture performed under anesthesia, and the animal was euthanized after the procedure by cervical dislocation. The ELISA test was performed in duplicate on each mouse, and IgG total (IgGT), IgG1, and IgG2a subclass antibody concentrations were determined [37].

The total number of attributes for this study was 67, and a database consisting of all the infected and control groups was created, during the acute and chronic *T. cruzi* infection phases, as shown in Table 1. It is included at the beginning of each variable in the list as an identification number (ID). The following section presents the proposed feature selection methods and classification by machine learning algorithms that have been considered and implemented in this work; then, a comparative discussion between obtained results in this study with other reported works is presented.

### 2.3. Multimodal Data Analysis

The proposed methodology for multimodal data analysis is presented in Figure 3. Initially, to observe the statistical relationships between all measured features, a general correlation map was calculated (considering infected and controls mice in acute and chronic stages). On the other hand, feature selection was conducted following four approaches: (1) empirical selection by an expert; (2) feature selection by Area Under the ROC curve (AUC-FS); (3) Random Forest (RF) and (4) Extremely Randomized Tree Classifier (ETC), where both RF and ETC used an impurity index; and (5) an ensemble voting process from all the features selected by the four previous approaches. Finally, since our goal is to establish a dichotomic classification, four automatic classification algorithms were implemented and validated: RF, ETC, Decision Trees (DT) and SVM. Data for classification was organized and processed as follows: acute phase (control + infected), chronic phase (control + infected), and general infection (acute phase + chronic phase).

**Table 1.** Murine experimental multimodal features in *Trypanosoma cruzi* infection.

| Diagnostic Modality | ID Extracted Feature | | | |
|---|---|---|---|---|
| ECG | 1. HR$^{+15,*8}$<br>2. HRV$^{+13,*7}$<br>3. CV$^{+14}$<br>4. RR Interval | 5. PQ Interval$^{*15}$<br>6. R Interval<br>7. QRS<br>8. QT Interval$^{+6,*9}$ | 9. ST Interval$^{+9,*10}$<br>10. QTc Interval<br>11. QTc$^{+11}$<br>12. QTc dispersion | 13. SR mean$^{+4}$<br>14. R Amplitude mean |
| ECHO | 15. HR SHORT AXIS<br>16. LVd$^{+5,*6}$ | 17. LVs<br>18. FS$^{*4}$ | 19. EF$^{*5}$ | |
| Doppler | 20. AbAO HR Avg<br>21. AbAO HR SD<br>22. AbAO RR internal Avg<br>23. AbAO RR interval SD<br>24. AbAO Peak velocity Avg$^{*16}$<br>25. AbAO Peak velocity SD<br>26. AbAO Minimum Flow Velocity Avg<br>27. AbAO Minimum Velocity SD<br>28. AbAO Mean Flow velocity Avg<br>29. AbAO Mean Flow velocity SD<br>30. AbAO Pulsability Index Avg<br>31. AbAO Pulsability Index SD | 32. AbAO Resistivity Index Avg<br>33. AbAO Resistivity Index SD<br>34. AO HR Avg<br>35. AO HR SD<br>36. AO RR Interval Avg<br>37. AO RR Interval SD<br>38. AO Pre-ejection time Avg$^{+8}$<br>39. AO Pre-ejection time SD<br>40. AO Peak velocity Avg$^{+11}$<br>41. AO Peak velocity SD<br>42. AO Stroke Distance Avg | 43. AO Stroke Distance SD<br>44. AO Ejection time Avg$^{*12}$<br>45. AO Ejection Time SD$^{+10}$<br>46. AO Rise Time Avg$^{*14}$<br>47. AO Rise Time SD<br>48. AO Mean velocity Avg<br>49. AO Mean velocity SD<br>50. AO Mean Acceleration Time Avg<br>51. AO Mean Acceleration Time SD<br>52. AO Peak Acceleration Avg$^{+7}$<br>53. AO Peak Acceleration SD | 54. MV HR Avg$^{+12}$<br>55. MV HR SD<br>56. MV RR Interval Avg<br>57. MV RR Interval SD<br>58. MV E Peak velocity Avg$^{*13}$<br>59. MV E Peak velocity SD<br>60. MV E Acceleration time Avg<br>61. MV E Acceleration time SD<br>62. MV E Peak to Peak time SD<br>63. MV E Deceleration time SD<br>64. MV E Deceleration Rate SD |
| ELISA | 65. IgGT$^{+1,*1}$ | 66. IgG1$^{+3,*3}$ | 67. IgG2a$^{+2,*2}$ | |

Doppler: AO (Aorta); AbAO (Abdominal Aorta); MV (Mitral Valve). $^{+n}$ Voting variables selection. $^{*n}$ Empirical variables selection. *n* indicates the relevance ranking.

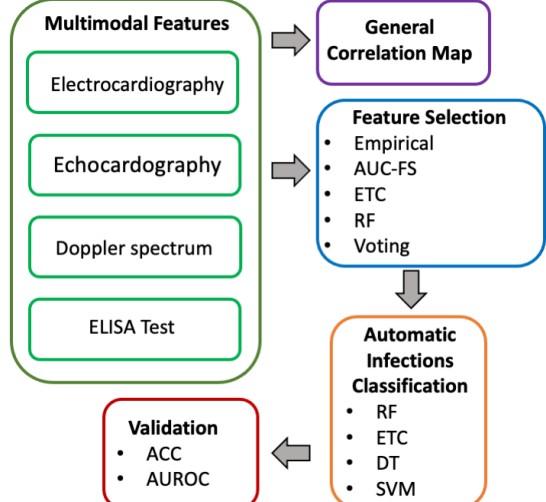

**Figure 3.** Proposed methodology diagram for multimodal data analysis. AUC-FS (Feature Selection by Area Under the Curve), ETC (Extremely Randomized Trees Classifier, RF (Random Forest), DT (Decision Tree), SVM (Support Vector Machine), ACC (Accuracy) and AUROC (Area Under the ROC Curve of the posterior probability.

### 2.3.1. Statistic Analysis

With the aim of unveiling the relationship between multimodal features, a Pearson correlation map was calculated. In this stage, all the variables measured from infected and controls mice were considered.

The correlation index detects linear correlations between variables of different modalities. In this type of analysis, a correlation of 1 indicates a complete dependency among variables, whereas a correlation of zero represents complete independence of the corresponding variables.

### 2.3.2. Feature Selection

Feature selection is considered a key stage when it comes to classification problems. In this work, a comparison between four different feature selection methods is presented: (a) an empirical selection; (b) a statistical method; (c) RF and (d) ETC machine learning techniques, and (d) general voting selection.

For the empirical selection, an expert in animal models for the *T. cruzi* disease selected, by descending relevance, a total of 16 features from the included variables in the experiment

based on her clinical expertise. In this approach, the best-ranked attributes are those that, according to the expert, best correlate with the disease stage.

From a statistical point of view, a Receiver Operating Characteristic (ROC) is a graphical plot of the fraction of True Positives (TP) vs. the fraction of False Positives (FP) for a binary classification system. The Area Under a ROC Curve (AUC) is a global measurement of the discrimination capabilities. AUC is also used to rank features and for feature selection [38]. In this work, a statistical analysis based on AUC is proposed to measure the individual importance of each considered variable in the study. The main objective of this approach is to select the statistically significant variables ($p < 0.05$) and rank them in relation to their AUC response. We refer to this approach as feature selection by AUC (AUC-FS), where each AUC-FS produces an improvement in the discrimination power of univariable classification. We applied AUC-FS to achieve a more accurate classification between control and infected groups.

In addition, based on machine learning, a feature selection analysis derived from the Mean Decrease in Impurity importance (MDI) [39,40] was implemented. This analysis aims to evaluate the importance of each variable by averaging the weighted impurity changes of all trees found by the corresponding methods. For this work, two MDIs were calculated: one based on RF and the other based on ETC. Both are learning algorithms widely used in classification and regression tasks, frequently applied for medical and biological data, given their learning capabilities and well-studied performance. These algorithms are based on multiple decision trees ensembles. The main difference between these two approaches consists in the data classification strategy: the RF algorithm considers an optimal segmentation of data, while the ETC algorithm considers that the separation planes are randomly selected. Another difference is the subsampling stage of the input data with replacement (boostrap) for RF and the complete data set processing for ETC. In this work, a total of 500 random trees and the GINI criterion for RF and ETC were used to calculate MDI (importance) for all analyzed features.

With the aim to identify which variables were selected by bith, the empirical method and automated-based approaches, namely AUC-FS, RF, and ETC algorithms, a general feature selection was carried out by voting, with the aim of identifying those that appear in 3 or 4 selection methods.

### 2.3.3. Automatic Classification

Four supervised classifiers were included for automatic classification of the *T. cruzi*-infected animals: RF, ETC, DT and SVM. As has been described, the first three classifiers are based on decision trees (classification model based on a logical decision graph); these classifiers have been widely used in various data medical or biological applications given their ability to learn from biological samples [41]. In contrast, the SVM classifier is a supervised learning algorithm that looks for a hyperplane that optimally separates classes using kernel functions [42]. For the RF, ETC and DT classifiers, the GINI impurity criterion was used considering the next parameters: maximum tree depth, minimum samples number required to split an internal node, minimum samples number required to be at a leaf node, and number of considered features to looking for the best split, which was optimized using a grid. Additionally, the optimization process for RF and ETC also considered the number of used trees. In the case of the SVM classifier, a radial-based kernel function was used, where the $C$ and *Gamma* parameters were also optimized with a grid.

### 2.3.4. Validation

The algorithms described in the previous subsection were used for binary classification of the three groups: acute phase (control vs. infected), chronic phase (control vs. infected), and general infection (acute + chronic). To observe the discrimination capability of each modality, each classifier was trained with the variables of each modality individually (ECG, ECHO, Doppler and ELISA). Afterward, for each classifier, the applied inputs consisted of all the multimodal variables described (67 features) and, in addition, the five subsets

of features previously selected by the empirical selection, AUC-FS, RF, ETC and general voting selection.

In all cases, a 5-cross-validation was carried out considering the 83.33% data for training and the 16.67% remaining data for a final unseen data validation. To evaluate the performance of each classifier, the accuracy (ACC) and the Area Under the ROC curve of the posterior probability (AUROC) were used. The ACC is a metric that measures the classification performance considering the number of correct predictions and the total amount of classified data. This metric can be defined in terms of positive and negative labels as:

$$ACC = \frac{TP + TN}{TP + FP + TN + FN} \tag{3}$$

where $TP$, $TN$, $FP$ and $FN$ are True Positives, True Negatives, False Positives, and False Negatives, respectively. The AUROC metric measures the posterior probability of each piece of classified data for a given class. It is a numeric value that represents the degree to which an instance is a member of a class. Both metrics (ACC and AUROC) range from 0 to 1, where 1 means a perfect classification.

The results and discussion of correlation maps, feature selection, and classification performance are shown in detail in the next section.

### 3. Results and Discussion

In order to confirm the evolution of the infection and study the stage of the disease, both parasitemia and mortality were recorded. During the acute phase, parasites were detected at day 15 post-infection (50,000–100,000 parasites/mL) and the peak of parasite was found at day 30 (0–3,450,000 parasites/mL) and dropping in count by day 35 (0–50,000 parasites/mL); the mortality rate was 24.2%. For the chronic phase, the peak of parasites was detected at day 30 (0–2,450,000 parasites/mL) and no parasites were detected on peripheral blood during days 60, 90 and 120; the mortality rate was over 50.8%. The parasitemia curve was similar to the results reported in other studies using the same strain of *T. cruzi* [28–30]. During the course of natural infection, the acute phase is characterized by parasites circulating in peripheral blood, where *T. cruzi* parasites travel to infect nucleated cells and replicate intracelularlly by binary fission. As the parasite has a tropism for cardiac muscular cells, it can no longer be found circulating in peripheral blood after the acute phase, and that is considered the beginning of the chronic phase. In the chronic phase, the parasite stays intracellularly, causing the development of different physiopathological mechanisms such as the immune-inflammatory response, autoimmunity, microvascular abnormalities, and nerve damage [43], all of them contribute to cardiac tissue damage, causing cardiomyopathy and sudden death in some cases.

Results from the univariate correlation analysis between all pairs of variables are shown below, where the correlation patterns between the different diagnostic modalities are displayed. Feature selection results are also presented for each one of the implemented methods (empirical, AUC-FS, RF, ETC, and voting selection). The selected features have been listed, according their relevance and contribution to an accurate discrimination between healthy and infected mice. We conclude this section showing the results of the implemented unimodal and multimodal classification approaches.

### 3.1. Pearson Correlation Maps—Statistic Analysis

To observe the relationship between variables of each modality, a Pearson correlation map was obtained (Figure 4). In this map, only correlations greater than or equal to 70% are presented. Each modality analyzed (ECG, ECHO, Doppler, and ELISA) is shown in a different line and bounding box color. An ID number has been added to each variable, which concurs with Table 1.

It can be seen that there are no correlations higher than 70% between modalities (blue square marks should be observed outside the bounding box of each modality). However,

there are significant correlations (r $\geq$ 70%) between variables of the same modality (blue square marks inside each bounding box). These results reflect linear independence between variables of different modalities; therefore, it is considered that each modality can provide different and complementary information and that a selection strategy of the best variables of each modality is necessary for the infection study and phase classification.

From a clinical point of view, among variables with high correlation, the HRV and CV parameters determine the cardiac cycle variability. Cardiac variability is a parameter to evaluate the function of the autonomic nervous system and is considered a predictor of sudden death for cardiac disease [44]. On the other hand, the high correlation between HR duration and other ECG parameters is well known. At higher HR rates, the duration values of the remaining parameters are reduced. The correlations between the ST and QTc intervals are related to ventricular repolarization abnormalities [45]. EF and FS parameters are calculated from LVd measurements during systole and diastole and, because of that, the ECHO variables studies are correlated. Other studies found a correlation between HR and systolic function parameters (EF and FS) and left ventricular wall motion, a parameter not included in the present study [25].

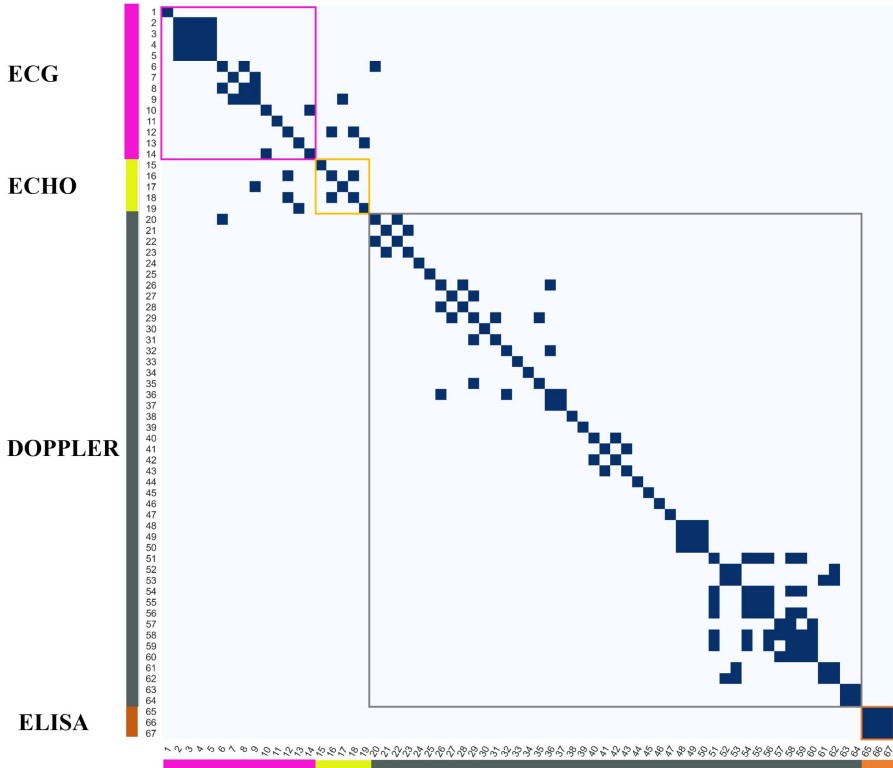

**Figure 4.** Univariable Pearson correlation map (r $\geq$ 0.70) for Multimodal Analysis for general *T. cruzi* Infection (ECG, ECHO, Doppler, and ELISA).

The results from the Doppler variables analysis show no correlations. However, alterations in blood flow of AbAO, AO, and MV were observed during the *T. cruzi* infection. They were mainly related to peak velocity and ejection time, both of which are components of the systolic wave produced during the contraction of the left ventricle, and blood flow volume is expelled through ascendant aorta AO and thereafter through the abdominal aorta AbAO. Changes related to a reduction in the blood vessel diameter produce a peak flow velocity rise. However, when a reduction in diameter is located distal to the site interrogated, the peak of velocity is reduced and a widening of the spectrum of the wave can be observed (rise in the pre-ejection time) [46]. Moreover, the infection with *T. cruzi* modifies the pulsatility and resistivity index of AbAO, which are parameters of peripheral resistance. The impact of *T. cruzi* on peripheral resistance has been poorly studied. The

results of the present study show us that peripheral resistance parameters should be considered during the study and diagnosis of *T. cruzi* infection.

The values of IgGT, IgG1, and IgG2a proved to be excellent predictors of infection in any of the stages of infection studied, where these predictors only indicate the presence of *T. cruzi* but not the stage of cardiac damage progression. The high prediction capabilities of these attributes is mainly explained by their associated immunological actions: IgG1 corresponds to an inflammatory response and IgG2a indicates anti-inflammatory activities [47].

### 3.2. Feature Selection

Table 1 indicates the most relevant characteristics according to the empirical selection (*), as well as the general selection by voting (+), where it can be seen that ELISA variables (IgGT, IgG1, IgG2a) have been considered the most relevant for both feature selection techniques. IgGT antibodies by ELISA are produced as a specific response against *T. cruzi*. These antibodies can be detected 2 to 3 weeks after infection and persist regardless of the stage of infection, making them a reliable test for the detection of parasite infection [47]. In the case of empirical selection, the importance order of diagnosis techniques was established as ELISA, ECHO, ECG, and Doppler; while, as a result of voting selection, the order of diagnosis techniques with more number of voted variables were ELISA, ECG, Doppler, and ECHO.

For the ECHO variables, only LVd was reported as relevant for both these feature selection methods.

In the case of ECG variables, empirical and voting selection show the QT interval, ST interval, HRV and HR as relevant; therefore, the assessment of cardiac electrical activity can be considered as a useful tool to identify infection, followed by abnormalities in ventricular repolarization (ST and QT), and the HRV and CV parameters denote changes in cardiac autonomic function.

In contrast, for the Doppler attributes, there are no features in common between the two selection methods, but in both cases, the majority of relevant features resulted from acquired variables of AO, AbAO and MV. These Doppler variables allow the identification of changes in the velocity of the flow in the blood vessels and valves, which could be associated with modifications in the diameter of the blood vessel and the resistance of peripheral blood vessels.

Figure 5 shows the selected features according to their relevance from the AUC-FS statistical analysis, as well as by the RF and ETC algorithms. In this figure, the graphics in red color are associated with the acute phase, the blue ones refer to the chronic phase, and the green color shows results in the case of the general infection. For the acute phase, 18 variables were obtained by AUC-FS statistical features selection, while from RF and ETC selection, 15 and 13 variables were obtained, respectively. For the chronic phase, 21 variables were obtained by AUC-FS selection, while from RF and ETC selection, 13 and 8 variables were obtained, respectively. Considering the general infection, 12 relevant variables were obtained by AUC-FS selection, for RF, only 8 variables were selected, and 10 variables were obtained by ETC selection. In these three cases (acute phase, chronic phase, and general infection), the ELISA variables (IgGT, IgG1, IgG2a) were considered as relevant features, where for the chronic phase and the general infection were the three variables with the highest relevance value. Further, in the acute phase, the relevant variables in common for these feature selection methods, are from ECG (QT and ST interval), Doppler (MV HR Avg), and ECHO (LVd); in the case of the chronic phase, for ECG (SR Mean, QTc, QT Interval) and Doppler (AO Ejection Time SD); and for general infection ECG (SR Mean), and Doppler (AO Pre ejection time AVG). Note that in the case of chronic phase and general infection, ECHO variables do not show high values of relevance.

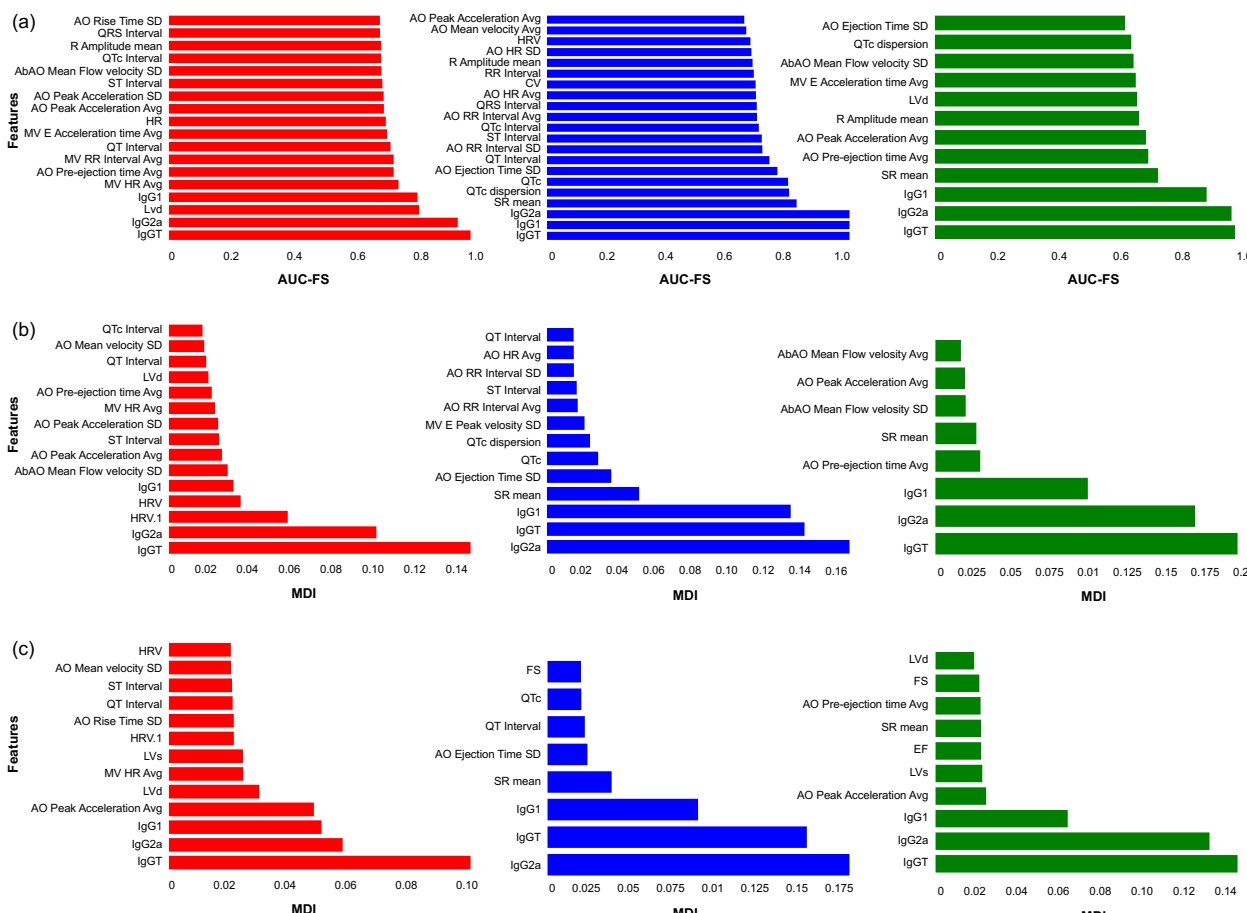

**Figure 5.** Multimodal Feature Selection Analysis for acute phase (red), chronic phase (blue), and general case of *T. cruzi* infection: (**a**) AUC-FS, (**b**) Random Forest, and (**c**) Extremely Randomized Trees.

During the acute phase of *T. cruzi* infection, the ELISA findings are the same as those for the general infection case. The predominance of IgG1 antibodies at this stage has been related to the presence of myocarditis [48]. Moreover, during this infection stage, the LVd diameter obtained by ECHO stands out. During *T. cruzi* infection, an early remodeling of the left ventricle occurs, where this change appeared to be characteristic for this stage. Moreover, abnormalities in ventricular repolarization reflected in QT and QTc during its passage through the Hiss bundle [49]. Additionally, the SR mean proves the early myocardial damage that occurs during this stage [50]. Finally, alterations in the AO peak acceleration time average parameter, which is the time between the start of systolic flow and the maximum velocity reached, is considered a non-invasive index of LV contractility in mice [51]. All these changes correspond to the relevant variables obtained by automatic feature selection algorithms.

During the chronic phase of infection, the production of IgG2a antibodies dominates over IgGT and IgG1. This finding concurs with other studies in mice [47]. This may be an indication that during chronic infection the predominant immune response is anti-inflammatory.

The findings regarding QT and QTc are repeated as in the case of acute infection, being able to conclude that once damage in ventricular repolarization appears, it will persist during the next stages of infection. The AO ejection time is described as the time lapse between the opening and closing of the aortic valve, and thus, it is a parameter related to the evaluation of ventricular function and ventricular contractility [52]. The relevant variables for the chronic phase of infection, obtained by the methods described in this

contribution, are rather similar to the reported clinical diagnosis and feature selection identified by machine learning techniques [24,25].

For the identification of general infection by *T. cruzi*, the detection of IgGT antibodies, as well as the IgG1 and IgG2a subtypes, is consistent with the predominance of an inflammatory immune response [47]. The Doppler evaluation of AO average time projection is a useful parameter, described as the time that elapses between ventricular depolarization and the onset of ventricular contraction while the heart valves are still closed; in addition, it can signify changes in myocardial contractility or, in the end, diastolic pressure of the ventricle [53]. Additionally, the SR variables, which may indicate myocardial damage [50], are consistent among the three classifiers. It is the same case with the acute and chronic phases of infection, the automatic feature selection is related to clinical damage reported for *T. cruzi* infection. It is remarkable that the variables obtained from the Doppler spectrum are relevant for CD diagnosis (acute phase, chronic phase and the general infection). Since the Doppler-related attributes are highly informative and the technique is non-invasive, those features may offer a potential tool to diagnose patients with CD.

### 3.3. Classification and Validation

The automatic classification metrics (ACC and AUROC), when using variables of each modality individually (ECG, ECHO, Doppler and ELISA), are presented in Table 2. This table shows the classification results for acute phase (control vs. infected), chronic phase (control vs. infected), and general infection groups (acute phase + chronic phase vs. controls). In each section, the results of the 5-fold cross-validation and final tests for the RF, ETC, DT and SVM classifiers are shown.

The results obtained in the classification between the Control vs. Acute groups show that ELISA modality has the best performance (around 90% for ACC and AUROC), followed by ECHO, ECG and Doppler (Table 2).

*T. cruzi* infection is presented in two stages: Acute and chronic. The presence and severity of clinical signs in each stage depends on the course of the infection on each animal. After the infection, IgG antibodies are produced during the first weeks of infection and remain present during the rest of the life of the infected organism. Therefore, ELISA tests appeared as the best diagnostic tools over the different clinical stages of the disease. Changes detected by the use of non-invasive modalities as ECHO appeared to be more frequent in the acute phase, if compared with alterations on the electrical conduction system of the heart detected by ECG and blood flow parameter changes diagnosed using Doppler in the experimental model studied.

Regarding the classification in the chronic phase (Control vs. Chronic groups Table 2), it can be observed that the ELISA test presents the best performance according to cross-validation (up to 100% for ACC and AUROC metrics), followed by Doppler modality (up to 80% for ACC and 85% for AUROC), ECG's variables (up to 70% for ACC and 75% for AUROC) and finally, ECHOs descriptors with performances up to 66% for ACC and 75% for AUROC. In contrast, in the final validation, it can be noted that ELISA and ECG modalities showed the best performances (up to 100%) in this classification.

The ELISA test proved to be the most useful indicator to detect infected animals at the chronic stage. On the other hand, changes detected by the sole use of Doppler were more frequent, if compared with the other modalities such as ECG and ECHO. As mentioned earlier, Doppler parameters should be taken into account during screening for *T. cruzi* infection diagnosis.

When focused on the general infection case (acute + chronic phase vs. controls classification), the results can be observed at the bottom of Table 2. It can be noted that ELISA tests show, again, the best performances, followed by ECG, ECHO and Doppler-related variables. The performances of the individual non-invasive methods were similar to the acute stage (up to 73% for ACC and up to 76% for AUROC during the cross-validation), which reflects the complexity of the classification task in these scenarios.

In a clinical scenario, ELISA tests for the detection of antibodies against *T. cruzi* are usually applied in patients when suspecting *T.cruzi* exposure. It has been demonstrated to be an excellent screening test. On the other hand, non-invasive diagnostic methods such as ECHO, ECG and Doppler are not specific tests that clearly indicate the exposure to the parasite, but they are useful to discern the stage of the disease as subjects become symptomatic. This elucidation is not possible with ELISA alone. Moreover, ECHO, ECG and Doppler methods are performed on patients suffering cardiac disease even when there is no suspicion of an infectious disease such as CD to be the cause of the cardiovascular disorders. It can be useful to consider *T. cruzi* infection based on a differential diagnosis considering alterations detected by these methods and include in the diagnosis whether there is or not a history of exposure to *T. cruzi* or if contact with the vector is suspected.

In general, the results presented in Table 2 show poor performance for variables obtained from non-invasive methods. Additionally, most of the results computed from the cross-validation present a large standard deviation (up to $\pm 30\%$ of the mean). Therefore, the combination of variables from different modalities and the incorporation of feature selection strategies are considered necessary for this study.

**Table 2.** Classification performance (%) for each modality: ECG, ECHO, Doppler and ELISA. Cross-validation results are expressed as mean and standard deviation. The best performances are presented in bold.

| Classifier | ECG | | ECHO | | Doppler | | ELISA | |
|---|---|---|---|---|---|---|---|---|
| | **ACC** | **AUROC** | **ACC** | **AUROC** | **ACC** | **AUROC** | **ACC** | **AUROC** |
| **Control vs. Acute, 5-Fold Cross-Validation (N = 30)** | | | | | | | | |
| RF | $53.3 \pm 22.1$ | $71.1 \pm 24.9$ | $\mathbf{70.0 \pm 12.5}$ | $66.7 \pm 14.1$ | $50.0 \pm 10.5$ | $\mathbf{68.9 \pm 11.4}$ | $90.0 \pm 13.3$ | $92.2 \pm 15.6$ |
| ETC | $63.3 \pm 16.3$ | $\mathbf{73.3 \pm 20.6}$ | $66.7 \pm 10.5$ | $66.7 \pm 12.2$ | $\mathbf{60.0 \pm 22.6}$ | $66.7 \pm 18.9$ | $90.0 \pm 13.3$ | $94.4 \pm 11.1$ |
| DT | $\mathbf{66.7 \pm 10.5}$ | $60.0 \pm 17.0$ | $\mathbf{70.0 \pm 12.5}$ | $73.3 \pm 8.2$ | $53.3 \pm 6.7$ | $53.3 \pm 6.7$ | $86.7 \pm 12.5$ | $93.3 \pm 13.3$ |
| SVM | $66.7 \pm 25.8$ | $71.1 \pm 23.9$ | $63.3 \pm 6.7$ | $\mathbf{82.2 \pm 8.9}$ | $50.0 \pm 23.6$ | $51.1 \pm 15.1$ | $\mathbf{90.0 \pm 8.2}$ | $\mathbf{95.6 \pm 5.2}$ |
| **Control vs. Acute, Final Test (N = 6)** | | | | | | | | |
| RF | $\mathbf{66.7}$ | $\mathbf{68.4}$ | $\mathbf{66.7}$ | $\mathbf{69.3}$ | $\mathbf{66.7}$ | $65.2$ | $\mathbf{100}$ | $\mathbf{98.5}$ |
| ETC | $\mathbf{66.7}$ | $67.3$ | $50.0$ | $52.1$ | $\mathbf{66.7}$ | $67.4$ | $\mathbf{100}$ | $98.4$ |
| DT | $50.0$ | $53.2$ | $\mathbf{66.7}$ | $65.2$ | $33.3$ | $40.3$ | $\mathbf{100}$ | $97.3$ |
| SVM | $50.0$ | $52.4$ | $50.0$ | $54.4$ | $\mathbf{66.7}$ | $\mathbf{70.2}$ | $83.3$ | $90.7$ |
| **Control vs. Chronic, 5-Fold Cross-Validation (N = 30)** | | | | | | | | |
| RF | $60.0 \pm 17.0$ | $\mathbf{75.6 \pm 16.3}$ | $56.7 \pm 22.6$ | $64.4 \pm 27.6$ | $73.3 \pm 13.3$ | $77.8 \pm 22.5$ | $\mathbf{100 \pm 0}$ | $\mathbf{100 \pm 0}$ |
| ETC | $63.3 \pm 16.3$ | $68.9 \pm 19.1$ | $56.7 \pm 22.6$ | $52.2 \pm 17.8$ | $\mathbf{80.0 \pm 6.7}$ | $\mathbf{85.6 \pm 10.9}$ | $\mathbf{100 \pm 0}$ | $\mathbf{100 \pm 0}$ |
| DT | $60.0 \pm 17.0$ | $56.7 \pm 17.0$ | $53.3 \pm 12.5$ | $46.7 \pm 12.5$ | $63.3 \pm 16.3$ | $66.7 \pm 18.3$ | $\mathbf{100 \pm 0}$ | $\mathbf{100 \pm 0}$ |
| SVM | $\mathbf{70.0 \pm 12.5}$ | $71.1 \pm 11.3$ | $\mathbf{66.7 \pm 14.9}$ | $\mathbf{75.6 \pm 13.0}$ | $76.7 \pm 13.3$ | $75.6 \pm 10.9$ | $95.7 \pm 6.3$ | $97.4 \pm 3.5$ |
| **Control vs. Chronic, Final Test (N = 6)** | | | | | | | | |
| RF | $\mathbf{100}$ | $\mathbf{100}$ | $50.0$ | $\mathbf{66.7}$ | $\mathbf{66.7}$ | $\mathbf{68.9}$ | $\mathbf{100}$ | $\mathbf{100}$ |
| ETC | $\mathbf{100}$ | $\mathbf{100}$ | $50.0$ | $58.9$ | $\mathbf{66.7}$ | $67.5$ | $\mathbf{100}$ | $\mathbf{100}$ |
| DT | $66.7$ | $67.8$ | $50.0$ | $52.4$ | $50.0$ | $54.4$ | $\mathbf{100}$ | $\mathbf{100}$ |
| SVM | $83.3$ | $88.9$ | $50.0$ | $53.4$ | $\mathbf{66.7}$ | $64.3$ | $\mathbf{100}$ | $\mathbf{100}$ |
| **Control vs. General Infection, 5-Fold Cross-Validation (N = 60)** | | | | | | | | |
| RF | $66.7 \pm 10.5$ | $\mathbf{72.9 \pm 8.9}$ | $\mathbf{73.3 \pm 9.7}$ | $72.1 \pm 9.0$ | $\mathbf{60.0 \pm 9.7}$ | $\mathbf{64.6 \pm 17.9}$ | $\mathbf{95.0 \pm 4.1}$ | $\mathbf{98.3 \pm 2.3}$ |
| ETC | $66.7 \pm 10.5$ | $69.9 \pm 12.7$ | $65.0 \pm 6.2$ | $67.7 \pm 9.7$ | $50.0 \pm 16.7$ | $54.6 \pm 16.2$ | $\mathbf{95.0 \pm 4.1}$ | $\mathbf{98.3 \pm 2.3}$ |
| DT | $\mathbf{66.7 \pm 9.1}$ | $64.6 \pm 9.3$ | $70.0 \pm 4.1$ | $69.6 \pm 6.4$ | $46.7 \pm 12.5$ | $45.0 \pm 8.5$ | $93.3 \pm 6.2$ | $93.0 \pm 6.4$ |
| SVM | $63.3 \pm 11.3$ | $69.4 \pm 18.2$ | $68.3 \pm 6.2$ | $\mathbf{76.0 \pm 5.9}$ | $50.0 \pm 9.1$ | $49.1 \pm 20.4$ | $86.7 \pm 8.5$ | $97.7 \pm 3.3$ |
| **Control vs. General Infection, Final Test (N = 12)** | | | | | | | | |
| RF | $\mathbf{83.3}$ | $\mathbf{75.7}$ | $\mathbf{58.3}$ | $45.7$ | $\mathbf{58.3}$ | $42.9$ | $\mathbf{83.3}$ | $85.7$ |
| ETC | $58.3$ | $71.4$ | $50.0$ | $47.2$ | $50.0$ | $41.4$ | $\mathbf{83.3}$ | $\mathbf{87.1}$ |
| DT | $50.0$ | $51.4$ | $\mathbf{50.0}$ | $\mathbf{51.4}$ | $50.0$ | $41.7$ | $\mathbf{83.3}$ | $82.9$ |
| SVM | $50.0$ | $52.3$ | $50.0$ | $51.3$ | $50.0$ | $45.7$ | $80.6$ | $85.7$ |

Automatic multimodal classification results for the acute phase (control vs. infected), chronic phase (control vs. infected), and general infection groups (acute phase + chronic phase vs. controls) are shown in Tables 3–5, respectively. In each table, the results of 5-fold cross-validation and the final tests for the RF, ETC, DT and SVM classifiers are presented. The columns contain the feature selection performance of each method as well as the result using all descriptors. The best performances for ACC and AUROC are highlighted in bold.

In the classification between mice with acute infection vs. control, it can be observed that the feature selection methods that we followed show better cross-validation perfor-

mance compared to the case in which all acquired features were considered. The best validation performances (ACC of 93.3 ± 13.3%) were obtained by combining the ETC classifier with RF, voting and ETC selections. Regarding AUROC, the combination of RF classifier/ETC selection and ETC classifier/RF selection showed the best performance (95.6 ± 8.9%). In the final test section, it can be seen that all approaches present good results, with RF selection together with ETC, DT and SVM classifiers being the combinations that showed an ACC and AUROC of 100%.

The classification results for mice with chronic infection vs. controls are presented in Table 4. In general, it can be observed that RF, ETC and DT classifiers show a perfect performance in combination with all sets of features used (except for the combination of the ETC classifier and RF selection). This behavior can be clearly noticed in the validation and final test with regard to ACC and AUROC metrics.

Regarding general infection (acute + chronic phase vs. controls), the classification results can be observed in Table 5. For the validation stage, it can be noted that the best obtained ACC was 96.7 ± 4.1%, considering any set of features selected considering the RF or ETC classifier. Regarding AUROC, the best obtained performance was 99.7 ± 0.6% for the ETC classifier with RF selection. In the final test, the RF and DT classifiers showed the best ACC with 83.3% for all sets of features used. For AUROC, the best performance was using classification and selection by ETC with 97.1%.

When comparing the results of the multimodal analysis (Tables 3–5) and the classification using variables of each modality individually (Table 2), it can be noted that the performance is generally consistent (mainly for the classification of acute stage vs. controls and general infection vs. controls). Furthermore, the standard deviation obtained during the cross-validation for the classification using multimodal variables is considerably lower compared to that shown in the unimodal classification. These results suggest that the combination and optimal selection of variables of a diverse nature can contribute to the adequate detection of *T. cruzi* infection.

As we already stated, the diagnosis of *T. cruzi* infection is complex. The acute phase of *T. cruzi* infection is short-lived, and the signs may not be clinically evident. If the acute infection is not treated, then a chronic condition is developed. In humans, diagnosis in the acute phase is critical because at this stage the treatment is most successful (between 74–89% efficient) [54]. According to the features evaluated in this contribution, the changes in the chronic phase are evident and easily identifiable with any of the used classifiers; however, the treatment applied during the chronic phase has a low success rate and the adverse effects outweigh the clinical benefit. Therefore, detection of cases in the acute phase would allow the identification and follow-up for adequate clinical management and timely treatment.

Comparison with Other Related Works and Clinical Diagnosis

Published works based on the ELISA test presented high accuracy in the classification of CD patients [24]. Therefore, the best results were achieved using an MLP algorithm presenting an ACC of 95.95%, 78.30% sensitivity, and specificity of 75.00% and AUROC of 0.861. Additional works reported relevant results, where arrhythmia classification from ECG variables by GDA- and SVM-based algorithms were able to discriminate six different types of cardiac arrhythmia: sinus rhythm, premature ventricular contraction, atrial fibrillation, sick sinus syndrome, ventricular fibrillation and 2nd-degree heart block. The results are outstanding as shown by the metrics such as ACC of 98.94%, 98.96%, 98.53%, 98.51%, 100% and 100%, respectively [27]. In addition, Escalera et al. [26] conducted a study regarding the classification of CD patients based on coronary damage and obtained an ACC of 72% for the case of three levels of damage in patients, using a high-resolution ECG and the ECOC algorithm. The attribute selection process led to the identification of the fractal short-term scaling exponent as the most prominent attribute [25].

According to our contribution, the multimodal features selection showed the ELISA test is the most relevant modality, followed by Doppler, ECG and ECHO in the case of the

general infection classification, where automatic classification showed up to 96.7% for ACC and 99.7% for AUROC during cross-validation, and as for the final test, performances of up to 83.3% for ACC and 97.1% for AUROC were obtained.

Unlike previously reported works, in this work we introduce the combination of four modalities. One of these modalities is the integration of Doppler-related attributes that have been only marginally studied for the *T. cruzi* infection model classification. Our results indicate that the Doppler test is a good cardiac damage descriptor for the infection stages and, together with the ECG and ECHO modalities, may serve as the basis for a non-invasive monitoring system.

**Table 3.** Multimodal classification performance (%) between control vs. acute infection groups. Cross-validation results are expressed as mean and standard deviation. The best performances are presented in bold.

| Classifier | All Features | | Empirical Selection | | AUC-FS Selection | | ETC Selection | | RF Selection | | Voting Selection | |
|---|---|---|---|---|---|---|---|---|---|---|---|---|
| | ACC | AUROC | ACC | AUROC | ACC | AUROC | ACC | AUROC | ACC | AUROC | ACC | AUROC |
| **5-Fold Cross-Validation (N = 30)** | | | | | | | | | | | | |
| RF | 90.0 ± 13.3 | 88.9 ± 22.2 | **93.3 ± 13.3** | 93.3 ± 13.3 | 90.0 ± 13.3 | 92.2 ± 15.6 | 90.0 ± 13.3 | **95.6 ± 8.9** | 90.0 ± 13.3 | 94.4 ± 11.1 | 93.3 ± 13.3 | 91.1 ± 17.8 |
| ETC | 86.7 ± 12.5 | 86.7 ± 21.5 | 90.0 ± 13.3 | 93.3 ± 13.3 | 86.7 ± 12.5 | 91.1 ± 17.8 | **93.3 ± 13.3** | 94.4 ± 11.1 | **93.3 ± 13.3** | **95.6 ± 8.9** | **93.3 ± 13.3** | 93.3 ± 13.3 |
| DT | 80.0 ± 12.5 | 73.3 ± 8.2 | 83.3 ± 14.9 | 76.7 ± 8.2 | 90.0 ± 13.3 | 86.7 ± 12.5 | 90.0 ± 13.3 | 86.7 ± 12.5 | 90.0 ± 13.3 | 83.3 ± 10.5 | 86.7 ± 12.5 | 90.0 ± 13.3 |
| SVM | 63.3 ± 24.5 | 75.6 ± 22.7 | 73.3 ± 22.6 | 80.0 ± 23.7 | 70.0 ± 28.7 | 82.2 ± 24.9 | 83.3 ± 18.3 | 91.1 ± 17.8 | 80.0 ± 24.5 | 84.4 ± 20.6 | 80.0 ± 24.5 | 82.2 ± 24.9 |
| **Final Test (N = 6)** | | | | | | | | | | | | |
| RF | 83.3 | 85.8 | 100 | 100 | 100 | 100 | 83.3 | 92.8 | 100 | 100 | 83.3 | 92.3 |
| ETC | 83.3 | 94.3 | 83.3 | 89.4 | 83.3 | 93.7 | 83.3 | 89.5 | 83.3 | 92.4 | 83.3 | 93.5 |
| DT | 100 | 100 | 100 | 100 | 100 | 100 | 100 | 100 | 100 | 100 | 100 | 100 |
| SVM | 100 | 100 | 83.3 | 91.3 | 83.3 | 92.4 | 100 | 100 | 100 | 100 | 66.7 | 72.5 |

**Table 4.** Multimodal classification performance (%) between control vs. chronic infection groups. Cross-validation results are expressed as mean and standard deviation. The best performances are presented in bold.

| Classifier | All Features | | Empirical Selection | | AUC-FS Selection | | ETC Selection | | RF Selection | | Voting Selection | |
|---|---|---|---|---|---|---|---|---|---|---|---|---|
| | ACC | AUROC | ACC | AUROC | ACC | AUROC | ACC | AUROC | ACC | AUROC | ACC | AUROC |
| **5-Fold Cross-Validation (N = 30)** | | | | | | | | | | | | |
| RF | 100 ± 0 | 100 ± 0 | 100 ± 0 | 100 ± 0 | 100 ± 0 | 100 ± 0 | 100 ± 0 | 100 ± 0 | 100 ± 0 | 100 ± 0 | 100 ± 0 | 100 ± 0 |
| ETC | 100 ± 0 | 100 ± 0 | 100 ± 0 | 100 ± 0 | 100 ± 0 | 100 ± 0 | 100 ± 0 | 100 ± 0 | 96.7 ± 6.7 | 98.4 ± 2.1 | 100 ± 0 | 100 ± 0 |
| DT | 100 ± 0 | 100 ± 0 | 100 ± 0 | 100 ± 0 | 100 ± 0 | 100 ± 0 | 100 ± 0 | 100 ± 0 | 100 ± 0 | 100 ± 0 | 100 ± 0 | 100 ± 0 |
| SVM | 90.0 ± 8.2 | 95.4 ± 2.5 | 96.7 ± 6.7 | 97.5 ± 7.4 | 93.3 ± 8.2 | 97.8 ± 4.4 | 90.0 ± 8.2 | 94.3 ± 4.6 | 93.3 ± 8.2 | 95.6 ± 3.2 | 96.7 ± 6.7 | 98.5 ± 3.4 |
| **Final Test (N = 6)** | | | | | | | | | | | | |
| RF | 100 | 100 | 100 | 100 | 100 | 100 | 100 | 100 | 100 | 100 | 100 | 100 |
| ETC | 100 | 100 | 100 | 100 | 100 | 100 | 100 | 100 | 100 | 100 | 100 | 100 |
| DT | 100 | 100 | 100 | 100 | 100 | 100 | 100 | 100 | 100 | 100 | 100 | 100 |
| SVM | 66.7 | 75.4 | 83.3 | 91.4 | 83.3 | 95.3 | 83.3 | 88.4 | 83.3 | 95.1 | 100 | 100 |

**Table 5.** Multimodal classification performance (%) between control vs. general infection (acute + chronic) groups. Cross-validation results are expressed as mean and standard deviation. The best performances are presented in bold.

| Classifier | All Features | | Empirical Selection | | AUC-FS Selection | | ETC Selection | | RF Selection | | Voting Selection | |
|---|---|---|---|---|---|---|---|---|---|---|---|---|
| | ACC | AUROC | ACC | AUROC | ACC | AUROC | ACC | AUROC | ACC | AUROC | ACC | AUROC |
| **5-Fold Cross-Validation (N = 60)** | | | | | | | | | | | | |
| RF | **96.7 ± 4.1** | 98.0 ± 2.8 | **96.7 ± 4.1** | 98.3 ± 2.3 | **96.7 ± 4.1** | 98.3 ± 2.3 | 95.0 ± 4.1 | 98.6 ± 1.8 | 95.0 ± 4.1 | 98.9 ± 2.3 | **96.7 ± 4.1** | 98.3 ± 2.2 |
| ETC | 90.0 ± 9.7 | 95.5 ± 5.3 | **96.7 ± 4.1** | 96.3 ± 5.5 | 95.0 ± 4.1 | 98.9 ± 1.4 | **96.7 ± 4.1** | 97.7 ± 3.3 | **96.7 ± 4.1** | **99.7 ± 0.6** | **96.7 ± 4.1** | 98.3 ± 2.3 |
| DT | 95.0 ± 4.1 | 93.0 ± 6.4 | 91.7 ± 7.5 | 91.3 ± 7.5 | 91.7 ± 5.3 | 93.0 ± 3.6 | 93.3 ± 3.3 | 93.0 ± 3.6 | 93.3 ± 3.3 | 93.0 ± 3.6 | 93.3 ± 6.2 | 91.3 ± 5.3 |
| SVM | 76.7 ± 12.2 | 93.8 ± 7.7 | 86.7 ± 8.5 | 94.4 ± 6.9 | 88.3 ± 8.5 | 95.5 ± 4.6 | 88.3 ± 6.7 | 96.6 ± 3.3 | 86.7 ± 8.5 | 97.7 ± 2.1 | 85.0 ± 9.7 | 98.3 ± 2.2 |
| **Final Test (N = 12)** | | | | | | | | | | | | |
| RF | **83.3** | 82.9 | **83.3** | 85.7 | **83.3** | 94.3 | **83.3** | 88.6 | **83.3** | 85.7 | **83.3** | 90 |
| ETC | **83.3** | 84.3 | 75 | 88.6 | **83.3** | 88.6 | 75 | **97.1** | 75 | 85.7 | 75 | 94.3 |
| DT | **83.3** | 85.7 | **83.3** | 85.7 | **83.3** | 85.7 | **83.3** | 85.7 | **83.3** | 85.7 | **83.3** | 85.7 |
| SVM | 66.7 | 74.3 | 66.7 | 74.3 | 66.7 | 82.9 | 66.7 | 71.4 | 58.3 | 71.4 | 75 | 82.9 |

## 4. Conclusions

In this work, we present a novel pipeline that integrates temporal data acquisition from four modalities, multiple feature selection analyses, and an automatic classification strategy of a *T. cruzi* infection model. The variables of a murine experimental model of *T. cruzi* infection were obtained from: ECG signals, ECHO images, Doppler spectrum, and ELISA antibody titers. We propose a unimodal analysis using correlation maps and a multimodal strategy using different feature selection approaches. Following this, a set of supervised classifiers were fed with different subsets of multimodal variables. The aim was to automatically classify *T. cruzi* infected animals from the murine experimental model for acute phase (control vs. infected), chronic phase (control vs. infected), and general infection groups (acute phase + chronic phase vs. controls).

The correlation map analysis of all 67 variables showed that there is no significant relationship between variables of different modalities; this suggests that the modalities analyzed show different and complementary information regarding immune response, electrical and cardiac mechanical functionality, and blood flow characteristics in the study of infection.

Regarding machine learning-based feature selection, we have shown that a handful of attributes and a strong, yet interpretable classifier, such as those in the decision tree family, are capable of identifying relevant changes along the stages in the *T. cruzi* murine infection model. The relevant attributes to tell apart *T. cruzi* infection (acute and chronic stages), as well as control animals, are variables acquired by ELISA (IgGT, IgG1, IgG2a), ECG (SR mean, QT Interval, ST Interval), Doppler (AO Peak Acceleration Avg, AO Pre-ejection time Avg, AO Ejection time SD), and ECHO (LVd).

The classification algorithms fed with different multimodal feature sets showed good performance in the discrimination of acute phase (control vs. infected), chronic phase (control vs. infected), and general infection groups (acute phase + chronic phase vs. control). In a clinical setting, one of the problems present in infected patients is the time they can remain asymptomatic (10–30 years), and thus, having an opportune early detection or an objective diagnostic method selection, even in symptomatic patients, is a great concern.

Some of the limitations presented in the research were the reduced number of animals studied due to the acquisition time needed per animal of the four modalities. As future work, we intend to include other variables extracted from ECHO (e.g., strain), ECG signals and/or other modalities such as histopathology that can be useful for a better understanding of the causes of the clinically detected alteration caused by the infection.

Our contribution is a step forward in the direction of providing medical specialists with an early detector of the Chagas disease. Since the path from the laboratory to the clinic is arduous and requires several validation stages, counting with a methodology that is able to detect physiological changes caused by the *T. cruzi* infection in animal models is of relevance.

Further studies are needed in order to determine the prognostic value for the cardiovascular disease evolution from the relevant variables found on each modality. It could be helpful for a better understanding of the pathophysiology of the *T. cruzi* infection and to improve the diagnosis and follow-up of the patients suffering from CD disease.

**Author Contributions:** Conceptualization, N.H.-M. and P.H.; data curation, N.H.-M., J.P.-G. and A.N.; formal analysis, N.H.-M., J.P.-G., A.N. and P.H.; funding acquisition, N.H.-M., J.P.-G., A.N. and P.H.; methodology, N.H.-M., J.P.-G. and P.H.; resources, N.H.-M., J.P.-G., A.N. and P.H.; validation, N.H.-M. and J.P.-G.; writing—original draft, N.H.-M., J.P.-G., A.N. and P.H.; writing—review and editing, N.H.-M., J.P.-G., A.N. and P.H. All authors have read and agreed to the published version of the manuscript.

**Funding:** This work was supported by UNAM-PAPIIT Programs IT100220, IA104622, IA103921, and CONACYT PDCPN-2015-102. The APC was funded by IT100220.

**Institutional Review Board Statement:** The experiment was approved by the ethical committee of the Centro de Investigaciones Regionales—Universidad Autónoma de Yucatán (CIRB-006-2017). The animals were handled according to the Guide for the Care and Use of Laboratory Animals (eighth edition) National Academies 2011.

**Informed Consent Statement:** Not applicable.

**Data Availability Statement:** The data used in this research can be shared via email.

**Acknowledgments:** The authors thank the Parasitology and Zoonoses Laboratories Staff of the Centro de Investigaciones Regionales Hideyo Noguchi at the Universidad Autonoma de Yucatan for help with data acquisition, and Luis J. Navarrete-Baduy for technical support.

**Conflicts of Interest:** The authors declare no conflict of interest.

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
