# Peer review of "Machine Learning-Based Feature Selection and Classification for the Experimental Diagnosis of Trypanosoma cruzi"

_electronics, doi:10.3390/electronics11050785_

Round 1

Reviewer 1 Report

  • The authors should describe the validity of the approach that how they validated the research work
  • Abstract and conclusion sections of the paper need to be revised in order to show the contributions of the study with the derivations of the study
  • What measures have been considered for evaluation for the research
  • English language need to be improved

Author Response

We greatly appreciate the comments and observations from the anonymous reviewers to improve the manuscript. We present this cover letter as a response to the reviewers’ observations and suggestions, and submit a revised version of the article with changes tracking.

Reviewer 1 

1.- The authors should describe the validity of the approach that how they validated the research work.

We thank the reviewer for this comment, which allowed us to describe in a clearer way our method. From a clinical point of view, the experiment consisting of T.cruzi infection on an animal model (mouse) was conducted in duplicate. Counting with a double independent experiment allowed us to observe replication in the data acquisition, which leads to a more robust analysis. We have modified the manuscript accordingly to include this information (lines 180-186). In order to verify that the infection was indeed present in the studied animals, we followed the described protocol to check for peripheral blood parasites by microscopy, as now described in the text (lines 200-201)

For data treatment, different features selectors and several machine learning algorithms were used to classify the animal subjects as described in subsections 2.3.2-2.3.4. The results obtained in the classification stage based on multimodal variables are consistent in both, the cross-validation, and the final test with hold-out one data item, as can be seen in Tables 3-5.

Additionally, a validation subsection has been added, which shows details of the classification strategy in the three study groups: acute phase (control vs. infected), chronic phase (control vs. infected), and general infection (acute + chronic). In addition, a more detailed description of the metrics used (ACC and AUROC) to evaluate the performance of the classifiers has been added (Lines 324-345).

2.- Abstract and conclusion sections of the paper need to be revised in order to show the contributions of the study with the derivations of the study.

The reviewer is completely right in his/her observation. The abstract and conclusions were not entirely comprehensive and clear. We used this opportunity to polish those sections as follows: The abstract (Lines 1-15) and conclusions (Lines 617-649) sections were extended, clarified, and complemented with comments from the two kind reviewers.

3.- What measures have been considered for evaluation for the research.

To emphasize the impact of a multimodal features acquisition and automatic classification algorithms, we included in section 2.2 a paragraph in the manuscript. 

Additionally, to quantitatively evaluate the performance of the classifiers trained with several subsets of multimodal features, ACC and AUROC were used as validation metrics. A validation subsection has been added describing in detail both metrics (Lines 324-345).

4.- English language need to be improved.

The English language of the manuscript was revised and modified under the supervision of a native speaker.

Reviewer 2 Report

Overall, this paper proposes a novel approach to detect T. cruzi infection by using multimodal diagnostic techniques and machine learning techniques.  The variables of a murine experimental model were acquired from ECG signals, ECHO images, Doppler, and ELISA. To classify the T.cuzi infected animals, the authors adopted four supervised classifiers, such as RF, ETC, DT, and SVM. The experimental results showed that the combination of four multimodal techniques with the combinations of RF, etc, and DT classifiers suggested high performance in terms of ACC and AUROC. 

(1) They need to address the limitations of the proposed method.
(2) Font sizes of figure 5 are too small. Please make them bigger and clear for readers.
(3) The manuscript requires careful proofreading.

This was well done and with these minor changes, should be ready for publication.

Author Response

We greatly appreciate the comments and observations from the anonymous reviewers to improve the manuscript. We present this cover letter as a response to the reviewers’ observations and suggestions, and submit a revised version of the article with changes tracking.

Reviewer 2

1.- They need to address the limitations of the proposed method.

We thank the reviewer for raising this issue. It gives us the opportunity to clearly show the main limitation of our otherwise, we believe, relevant contribution. The main limitations of the work are related to the number of animal models studied and the acquisition time per day of the four modalities described. We have made these limitations clear in the text by adding this information in the conclusions section (Lines 639-649).

2.- Font sizes of figure 5 are too small. Please make them bigger and clear for readers.

Once again, we thank the reviewer for pointing out this issue. Figure 5 was modified according to reviewer’s suggestion. The font size of figure 5 has been increased (Page 13).

3.- The manuscript requires careful proofreading.

The English language of the manuscript was revised.
